# A conformational fingerprint for amyloidogenic light chains

**Cristina Paissoni[1†], Sarita Puri[1,2†], Luca Broggini[3], Manoj K Sriramoju[4], Martina Maritan[1], Rosaria Russo[5], Valentina Speranzini[1], Federico Ballabio[1], Mario Nuvolone[6,7], Giampaolo Merlini[6,7], Giovanni Palladini[6,7], Shang-Te Danny Hsu[4,8,9], Stefano Ricagno[1,3]\*, Carlo Camilloni[1]\***

[1]Department of Bioscience, University of Milan, Milan, Italy; [2]Indian Institute of Science Education and Research Pune, Pune, India; [3]Institute of Molecular and Translational Cardiology, IRCCS, Policlinico San Donato, Milan, Italy; [4]Institute of Biological Chemistry, Academia Sinica, Taipei, Taiwan; [5]Department of Pathophysiology and Transplantation, University of Milan, Milan, Italy; [6]Department of Molecular Medicine, University of Pavia, Pavia, Italy; [7]Amyloidosis Research and Treatment Center, Fondazione IRCCS Policlinico San Matteo, Pavia, Italy; [8]Institute of Biochemical Sciences, National Taiwan University, Taipei, Taiwan; [9]International Institute for Sustainability with Knotted Chiral Meta Matter (SKCM[2]), Hiroshima University, Higashi-Hiroshima, Japan

**\*For correspondence:**
stefano.ricagno@unimi.it (SR);
carlo.camilloni@unimi.it (CC)

[†]These authors contributed equally to this work

**Competing interest:** The authors declare that no competing interests exist.

## eLife Assessment

This study addresses an **important** and long-standing question regarding the molecular mechanism of protein misfolding in Ig light-chain (LC) amyloidosis (AL), a life-threatening condition. By combining advanced techniques, including small-angle X-ray scattering, molecular dynamics simulations, and hydrogen-deuterium exchange mass spectrometry, the authors provide **convincing** evidence that the 'H state' distinguishes amyloidogenic from non-amyloidogenic LCs. These findings not only offer novel insights into LC structural dynamics but also hold promise for guiding therapeutic strategies in AL and will be of particular interest to structural biologists, biophysicists, and many others working on amyloid diseases.

**Abstract** Both immunoglobulin light-chain (LC) amyloidosis (AL) and multiple myeloma (MM) share the overproduction of a clonal LC. However, while LCs in MM remain soluble in circulation, AL LCs misfold into toxic-soluble species and amyloid fibrils that accumulate in organs, leading to distinct clinical manifestations. The significant sequence variability of LCs has hindered the understanding of the mechanisms driving LC aggregation. Nevertheless, emerging biochemical properties, including dimer stability, conformational dynamics, and proteolysis susceptibility, distinguish AL LCs from those in MM under native conditions. This study aimed to identify a[2] conformational fingerprint distinguishing AL from MM LCs. Using small-angle X-ray scattering (SAXS) under native conditions, we analyzed four AL and two MM LCs. We observed that AL LCs exhibited a slightly larger radius of gyration and greater deviations from X-ray crystallography-determined or predicted structures, reflecting enhanced conformational dynamics. SAXS data, integrated with molecular dynamics simulations, revealed a conformational ensemble where LCs adopt multiple states, with variable and constant domains either bent or straight. AL LCs displayed a distinct, low-populated, straight conformation (termed H state), which maximized solvent accessibility at the interface between constant and variable domains. Hydrogen-deuterium exchange mass spectrometry experimentally

validated this H state. These findings reconcile diverse experimental observations and provide a precise structural target for future drug design efforts.

## Introduction

Immunoglobulin light-chain (LC) amyloidosis (AL) is a systemic disease associated with the overproduction and subsequent amyloid aggregation of patient-specific LCs (*Merlini et al., 2018*; *Blancas-Mejia et al., 2018*; *Cascino et al., 2022*; *Poshusta et al., 2009*). Such aggregation may take place in one or several organs, the heart and kidneys being the most affected ones (*Merlini et al., 2018*). AL originates from an abnormal proliferation of a plasma cell clone that results in LCs' overexpression and oversecretion in the bloodstream (*Merlini et al., 2018*). LCs belonging to both $\lambda$ and κ isotypes are associated with AL; however, $\lambda$-LCs are greatly over-represented in the repertoire of AL patients. Specifically, AL-causing LCs (AL-LCs) most often belong to a specific subset of $\lambda$ germlines such as *IGLV6* ($\lambda$6), *IGLV1* ($\lambda$1), and *IGLV3* ($\lambda$3) (*Perfetti et al., 2012*; *Kourelis et al., 2017*; *Comenzo et al., 2001*; *Absmeier et al., 2023*).

$\lambda$-LCs are dimeric in solution, with each subunit characterized by two immunoglobulin domains, a constant domain (CL) with a highly conserved sequence and a variable domain (VL) whose extreme sequence variability is the result of genomic recombination and somatic mutations (*Bourne et al., 2002*; *Chiu et al., 2019*; *Di Noia and Neuberger, 2007*; *Oberti et al., 2017*). VL domains are generally indicated as the key responsible for LC amyloidogenic behavior. The observation that the fibrillar core in most of the structures of ex vivo AL amyloid fibrils consists of VL residues further strengthens this hypothesis (*Radamaker et al., 2021a*; *Radamaker et al., 2021b*; *Radamaker et al., 2019*; *Swuec et al., 2019*; *Puri et al., 2023*). However, in a recent cryo-electron microscopy structure, a stretch of residues belonging to the CL domain is also part of the fibrillar core, and mass spectrometry (MS) analysis of several ex vivo fibrils from different patients indicates that amyloids are composed of several LC proteoforms, including full-length LCs (*Schulte et al., 2024*; *Lavatelli et al., 2020*; *Mazzini et al., 2022*; *Dasari et al., 2015*).

Interestingly, the overproduction of an LC is a necessary but not sufficient condition for the onset of AL. Indeed, the uncontrolled production of a clonal LC is often associated with multiple myeloma (MM), a blood cancer, but only a subset of MM patients develops AL, thus indicating that specific sequence/biophysical properties determine LC amyloidogenicity and AL onset (*Oberti et al., 2017*; *Rottenaicher et al., 2021*; *Rennella et al., 2019*; *Klimtchuk et al., 2023*). To date, the extreme sequence variability of AL-LCs has prevented the identification of sequence patterns predictive of LC amyloidogenicity; however, it has been reproducibly reported that several biophysical properties correlate with LC aggregation propensity. AL-LCs display a lower thermodynamic and kinetic fold stability compared with non-amyloidogenic LCs found overexpressed in MM patients (hereafter MM-LCs) (*Oberti et al., 2017*; *Mazzini et al., 2022*; *Dasari et al., 2015*; *Rottenaicher et al., 2021*; *Rennella et al., 2019*; *Klimtchuk et al., 2023*).

Previous work on LCs has indicated how differences in conformational dynamics can play a role in the aggregation properties of AL-LCs (*Rottenaicher et al., 2021*; *Rennella et al., 2019*; *Klimtchuk et al., 2023*; *Weber et al., 2018*; *Sun et al., 2023*). Oberti et al. compared multiple $\lambda$-LCs obtained from either AL patients or MM patients identifying the susceptibility to proteolysis as the best biophysical parameter distinguishing the two sets (*Oberti et al., 2017*). Weber et al. showed, using a murine-derived κ-LC, how a modification in the linker region can lead to a greater conformational dynamic, an increased susceptibility to proteolysis, as well as an increased in vitro aggregation propensity (*Weber et al., 2018*). Additionally, AL-LC flexibility and conformational freedom have also been correlated to the proteotoxicity observed in patients affected by cardiac AL and experimentally verified in human cardiac cells and a *Caenorhabditis elegans* model (*Maritan et al., 2020*; *Broggini et al., 2023*). It is noteworthy that the amyloid LCs analyzed in this study were originally purified from patients with cardiac AL.

Here, building on this previous work as well as on our previous experience on β2-microglobulin, another natively folded amyloidogenic protein (*Visconti et al., 2019*; *Sala et al., 2020*; *Camilloni et al., 2016*; *Le Marchand et al., 2018*; *Achour et al., 2020*), we aimed at revealing the structural and dynamic determinants of LC amyloidogenicity. With this goal, we investigated the native solution-state dynamics of multiple $\lambda$-LCs by combining MD simulations, small-angle X-ray scattering (SAXS),

**Table 1.** LC systems studied in this work.

The table includes information about the germline, phenotype, method to obtain structure, the SAXS curves, and the radius of gyration derived from the SAXS data for all the model proteins studied in this work.

| LC | Germline | Phenotype | Structure | SAXS $\chi^2$ q < 0.5 Å (q < 0.3) | $R_g$ (SAXS) (nm) |
|---|---|---|---|---|---|
| H3 | IGLV1-44*01 | AL | 5MTL | 1.6 (1.9) | 2.57 ± 0.02 |
| H7 | IGLV1-51*01 | AL | 5MUH | 2.8 (4.0) | 2.56 ± 0.02 |
| H18 | IGLV3-19*01 | AL | Homology | 1.6 (1.9) | 2.56 ± 0.01 |
| AL55 | IGLV6-57*02 | AL | Homology | 5.1 (7.8) | 2.58 ± 0.04 |
| M7 | IGLV3-19*01 | MM | 5MVG | 1.2 (1.2) | 2.50 ± 0.02 |
| M10 | IGLV2-14*03 | MM | AF2 | 1.2 (1.2) | 2.51 ± 0.01 |

AL, amyloidosis; LC, light chain; MM, multiple myeloma; SAXS, small-angle X-ray scattering.

and hydrogen-deuterium exchange mass spectrometry (HDX-MS). Interestingly, we found a unique conformational fingerprint of amyloidogenic LCs corresponding to a low-populated state (defined as H state hereon) characterized by extended linkers, with an accessible VL-CL interface and possible structural rearrangements in the CL-CL interface.

## Results

### SAXS suggests differences in the conformational dynamics of amyloidogenic and non-amyloidogenic LC

SAXS data were acquired either in bulk or in line with size-exclusion chromatography (SEC) for a set of six LCs previously described (*Table 1* and 'Materials and methods'). Four of these LCs (referred to as H3, H7, and H18 in *Oberti et al., 2017* and AL55 in *Swuec et al., 2019*) were identified in multiple AL patients (AL-LC), while two (referred to as M7 and M10 in *Oberti et al., 2017*) were identified in MM patients (MM-LC). These LCs cover multiple germlines, with H18 and M7 belonging to the same germline (*Table 1*). The sequence identity is the largest for H18 and M7 (91.6%), while it is the lowest for AL55 and M7 (75.2%). A table showing the statistics for all pairwise sequence alignments is provided in *Supplementary file 1a*. For H3, H7, and M7, a crystal structure was previously determined (*Oberti et al., 2017*), while for H18, AL55, and M10, we obtained a model using either homology modeling (H18 and AL55) or AlphaFold2 (M10). Qualitatively, the SAXS curves in *Figure 1* did not reveal any macroscopic deviation of the solution behavior with respect to the crystal or model conformation. For each LC, we compared the experimental and theoretical curves calculated from the LC structures (*Table 1*) analyzing the residuals and the associated $\chi^2$. The analysis indicated a discrepancy between the model conformation and the data in the case of the AL-LCs, which was instead not observed in the case of the MM-LCs. For AL-LC, residuals deviate from normality in the low q region, suggesting some variability in the global size of the system. Additionally, a weak trend distinguishing AL-LC from MM-LCs could be identified in the radius of gyration ($R_g$) as derived from the Guinier approximation (*Table 1*). H3, H7, H18, and AL55 have an $R_g$ that is 0.5–0.8 Å larger than M7 and M10, a small but statistically significant difference (p-value<$10^{-5}$). Overall, the SAXS measurements point to less compact and more structurally heterogeneous AL-LCs compared with more compact and structurally homogeneous MM-LCs.

### MD simulations reveal a conformational fingerprint for amyloidogenic light chains

To investigate the conformational dynamics of the six LCs, we performed metadynamics metainference (M&M) MD simulations employing the SAXS curves (q < 0.3 Å) as restraints ('Materials and methods'; *Bonomi et al., 2016b*; *Bonomi et al., 2016a*; *Paissoni et al., 2020*; *Paissoni and Camilloni, 2021*). Metainference is a Bayesian framework that allows the integration of experimental knowledge on-the-fly in MD simulations, improving the latter while accounting for the uncertainty in the data and their interpretation. Metadynamics is an enhanced sampling technique that is able to speed

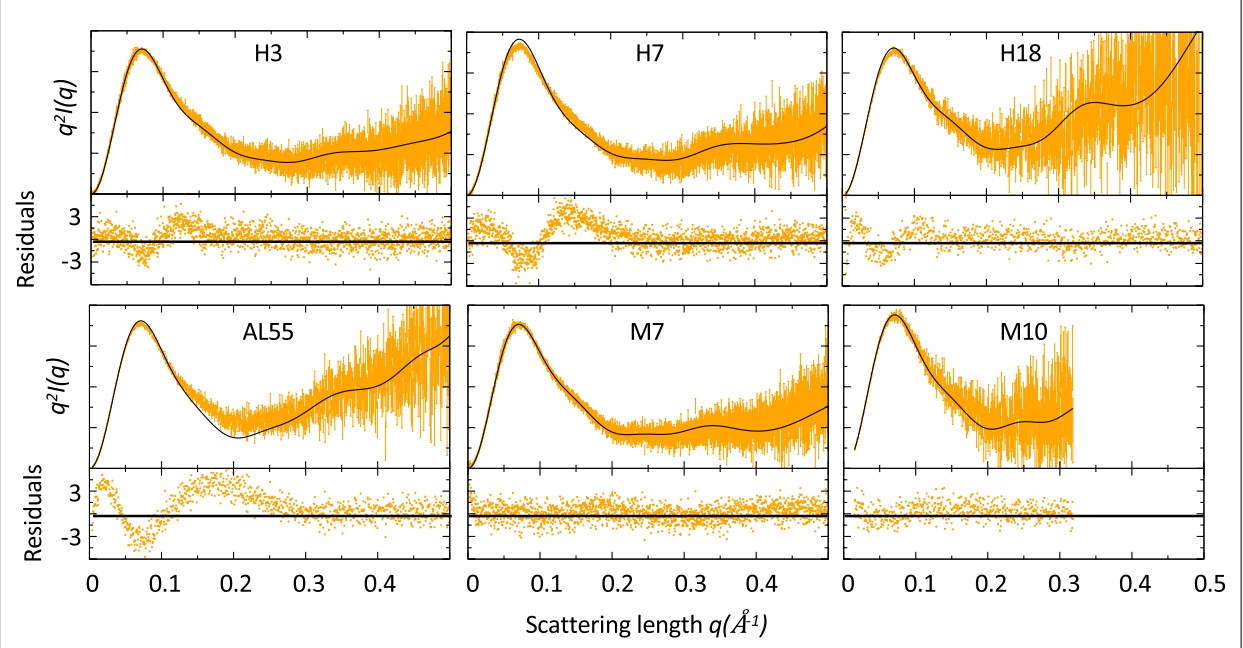

**Figure 1.** Comparison of small-angle X-ray scattering (SAXS) data and single light-chain (LC) structures. Kratky plots of for amyloidosis (AL) and multiple myeloma (MM) LCs. The experimental (orange) and theoretical (black) curves (top panels) and the associated residuals (bottom panels) indicate that AL-LC solution behavior deviates from reference structures more than MM-LC. SAXS was measured as follows: H3 measured in bulk (Hamburg), 3.4 mg/ml; H7 measured in bulk (Hamburg), 3.4 mg/ml; H18 measured by size-exclusion chromatography (SEC)-SAXS (ESRF [European Synchrotron radiation facility]) with the injection concentration of at 2.8 mg/ml; AL55 measured in bulk (ESRF), 2.6 mg/ml; M7 measured in bulk (Hamburg), 3.6 mg/ml; and M10 measured by SEC-SAXS with the injection concentration of 6.7 mg/ml (ESRF). Theoretical SAXS curves were calculated using crysol (*Manalastas-Cantos et al., 2021*). Log-log plots are shown in *Figure 1—figure supplement 1*.

The online version of this article includes the following figure supplement(s) for figure 1:

**Figure supplement 1.** Log-log plots representing the small-angle X-ray scattering experimental curves for the six proteins.

**Figure supplement 2.** Size exclusion coupled multiangle light scattering analysis of purified light chains (AL55, H3, H7, H18, M7, and M10).

**Figure supplement 3.** Small-angle X-ray scattering–size-exclusion chromatography average intensity for H18 and M10.

up the sampling of the conformational space of complex systems. The combination of SAXS and MD simulations has been shown to be effective for multidomain proteins, as well as for intrinsically disordered proteins (*Saad et al., 2021*; *Ahmed et al., 2021*; *Thomasen and Lindorff-Larsen, 2022*).

For each LC, we performed two independent M&M simulations coupled by the SAXS restraint, accumulating around 120–180 µs of MD per protein ('Materials and methods' and *Table 2*). The resulting conformational ensembles resulted in a generally improved agreement with the SAXS data employed as restraints (*Table 2* and *Figure 2A*). To investigate the differences in the LC local flexibility, we first analyzed the root mean square fluctuations (RMSFs) for the CL and VL separately, averaging over the chains and the replicates (*Figure 2B*). The RMSF indicates comparable flexibility in most of the regions, with differences localized in the termini and in some loops. The VLs of amyloidogenic LCs are generally more flexible than the MM ones, but this may be associated with the lengths of their complementarity-determining regions (CDRs). Indeed, M10 has the longest and most flexible CDR1 and the shortest and least flexible CDR3. Unexpectedly, there are some differences also in the CL domains. Here, in *Figure 2B*, the AL-LCs are always more flexible in at least one region even if these differences are relatively small. Overall, the RMSF does not provide a clear indication to differentiate AL and MM-LCs. To provide a global description of the dynamics of the six LC systems, we then introduced two collective variables, namely the elbow angle, describing the relative orientation of VL and CL dimers, and the distance between the VL and CL dimers center of mass, illustrated in *Figure 2C*.

In *Figure 3*, we report the free energy surfaces (FESs) obtained from the processing of the two replicates of each LC as a function of the elbow angle and the CL-VL distance calculated from their center of mass. The visual inspection of the FES indicates converged simulation: in all cases, the replicates explore a comparable free-energy landscape with comparable features. All six LC FESs

**Table 2.** Metainference simulations performed in this work for the six systems.

For each metainference simulation, the simulation time per replica with the number of replicas is reported; the $\chi^2$ of the resulting conformational ensemble with the experimental SAXS curve, the range $q < 0.3$ Å is the one used as restraint in the simulation; the average radius of gyration with error estimated by block averaging.

| LC code | Simulation | Length per replica (ns) (# replicas) | SAXS $\chi^2$ $q < 0.5$ Å ($q < 0.3$) | Average $R_g$ (nm) |
|---|---|---|---|---|
|  | M&M 1 | 1530 (60) | 1.2 (1.1) | 2.56 ± 0.02 |
| H3 | M&M 2 | 1520 (60) | 1.2 (1.1) | 2.56 ± 0.02 |
|  | M&M 1 | 1627 (60) | 1.1 (1.2) | 2.54 ± 0.04 |
| H7 | M&M 2 | 1545 (60) | 1.1 (1.2) | 2.54 ± 0.05 |
|  | M&M 1 | 1643 (60) | 1.4 (1.6) | 2.53 ± 0.03 |
| H18 | M&M 2 | 1529 (60) | 1.4 (1.7) | 2.54 ± 0.03 |
|  | M&M 1 | 1545 (60) | 2.7 (3.0) | 2.59 ± 0.02 |
| AL55 | M&M 2 | 1591 (60) | 2.5 (2.7) | 2.58 ± 0.05 |
|  | M&M 1 | 1623 (60) | 1.2 (1.2) | 2.52 ± 0.03 |
| M7 | M&M 2 | 1530 (60) | 1.1 (1.2) | 2.52 ± 0.03 |
|  | M&M 1 | 987 (60) | 1.1 (1.1) | 2.53 ± 0.07 |
| M10 | M&M 2 | 995 (60) | 1.1 (1.1) | 2.54 ± 0.05 |

LC, light chain; M&M, metadynamics metainference; SAXS, small-angle X-ray scattering.

share common features: a relatively continuous low free energy region along the diagonal, spanning configurations where the CL and VL are bent and close to each other (state $L_B$), and configurations where the CL and VL domains are straight and at relative distance between 3.4 and 4.1 nm (state $L_S$). A subset of LCs, namely H18, M7, and AL55, display conformations where the domains are straight in line (elbow angle >2.5 rad) and in close vicinity, with a relative distance between the center of mass of <3.4 nm (state G). Of note, H18 and M7 belong to the same germline, letting us speculate that this state G may be germline specific. Most importantly, only the AL-LCs display configurations with CL and VL straight in line but well separated at relative distances >4.1 nm; this state H seems to be a fingerprint specific for AL-LCs. A set of configurations exemplifying the four states is reported in *Figure 3*. The estimates of the populations for the four states $L_B$, $L_S$, G, and H are reported in *Table 3*. The quantitative analysis indicates that, within the statistical significance of the simulations, states $L_B$ and $L_S$ represent in all cases most of the conformational space. In the case of H18, AL55, and M7, the compact state G is also significantly populated (10–34%). The state H, associated with amyloidogenic LCs, is populated between 5 and 10% in H3, H7, H18, and AL55 and <1% in M7 and M10.

To identify additional differences between the conformations observed in state H and the rest of the conformational space, we focused our attention on the VL-VL and CL-CL dimerization interfaces. In *Figure 4*, we showed the free energy as a function of the distance between the CL domains versus the distance between the VL domains for each of the four states for one of the two simulations performed on H3; the same analysis for all other simulations is shown in *Figure 4—figure supplements 1–6*. From the comparison of the FESs, it is clear that only in the conformations corresponding to the state H do the CL-CL dimers display an alternative configuration. In the case of H3, the CL-CL domains in the H state are characterized by a shift toward the configurations characterized by a larger distance; the same is observed in the case of H18 and AL55, while in the case of H7, the H state is characterized by a smaller distance between the CL domains.

Our conformational ensembles allowed us to hypothesize a conformational fingerprint for AL proteins, namely the presence of a weakly but significantly populated state (H) characterized by a more extended quaternary structure, with VL and CL dimers well separated, and with perturbed CL-CL interfaces.

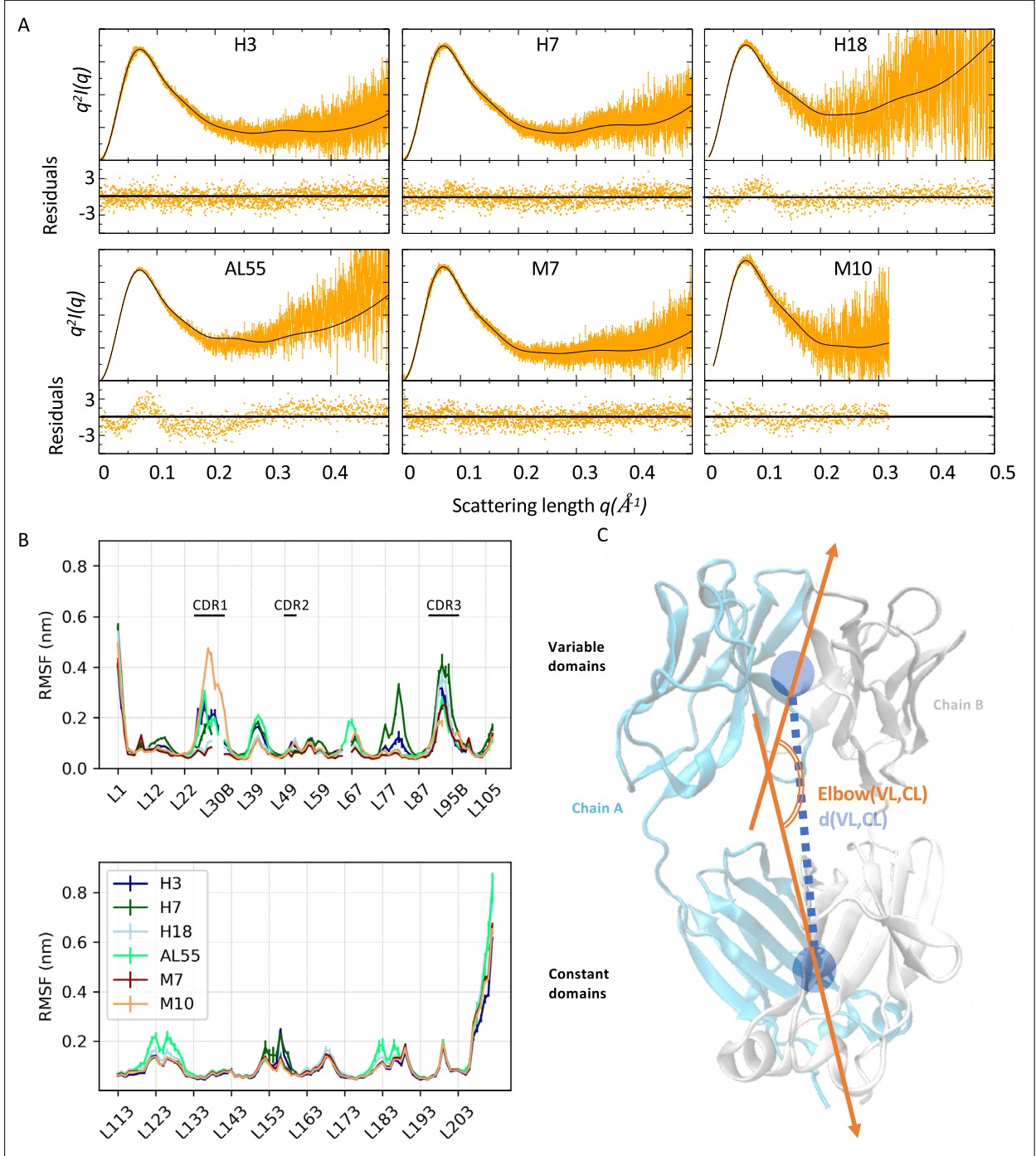

**Figure 2.** Light-chain small-angle X-ray scattering (SAXS)-driven molecular dynamics simulations. (**A**) Kratky plots and associated residuals (bottom panels) comparing experimental (orange), and theoretical (black) curves obtained by averaging over the metainference ensemble for H3, H7, H18, AL55, M7, and M10, respectively. Theoretical SAXS curves were calculated using crysol (*Manalastas-Cantos et al., 2021*). (**B**) Residue-wise root mean square fluctuations (RMSFs) obtained by averaging the two metainference replicates and the two equivalent domains for the six systems studied. The top panel shows data for the variable domains, while the bottom panel shows data for the constant domain. Residues are reported using *Chothia and Lesk* numbering (*Al-Lazikani et al., 1997*). (**C**) Schematic representation of two global collective variables used to compare the conformational dynamics of the different systems, namely the distance between the center of mass of the variable domain (VL) and constant domain (CL) dimers and the angle describing the bending of the two domain dimers.

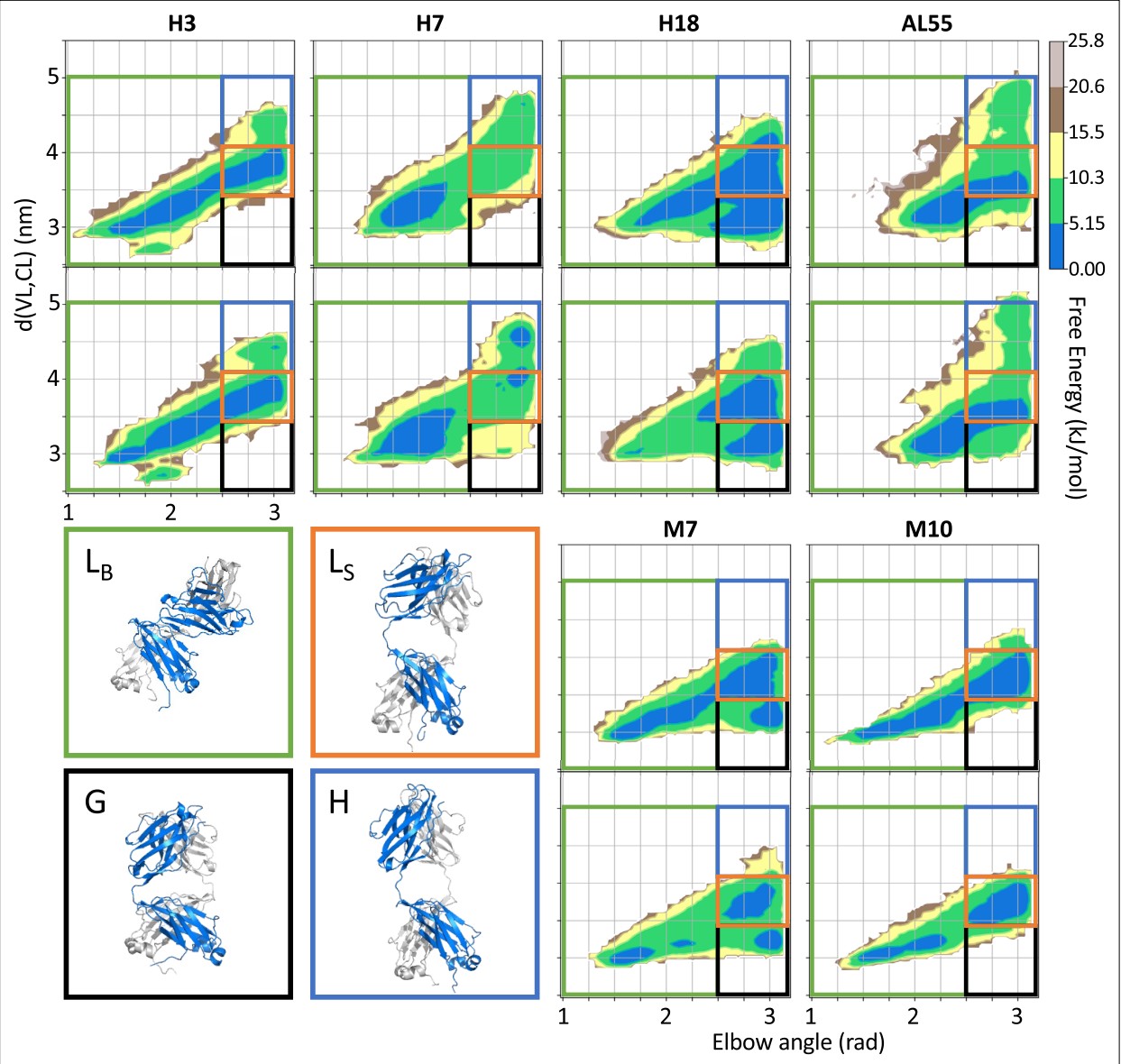

**Figure 3.** Free energy surfaces (FESs) for the six light-chain systems under study by metadynamics metainference molecular dynamics simulations. For each system, the simulations are performed in duplicate. The x-axis represents the elbow angle, indicating the relative bending of the constant and variable domains (in radians), while the y-axis represents the distance in nm between the center of mass of the constant domain (CL) and variable domain (VL) dimers. The free energy is shown with color and isolines every $2k_BT$ corresponding to 5.16 kJ/mol. On each FES are represented four regions (green, $L_B$ state; red, $L_S$ state; blue, H state; and black, G state), highlighting the main conformational states. For each region, a representative structure is reported in a rectangle of the same color.

## HDX independently validates the amyloidogenic LC conformational fingerprint

To gain further molecular insight into how the dynamics of the tertiary and quaternary structures can be differentiated in AL- and MM-LCs, HDX-MS was performed on our set of proteins. HDX-MS probes the protein dynamics by monitoring the hydrogen-to-deuterium uptake over time and the obtained data well complement structural, biophysical, and computational data. Four LCs from our set (H3, H7, AL55, and M10) yielded good peptide sequence coverages of 98.6, 92.5, 98.6, and 99.1%, respectively, with a redundancy of >4.0 (*Figure 5—figure supplements 1–4* and *Supplementary file 1b*). H18 and M7 were not included in this analysis due to their poor sequence coverage and were not further investigated. Due to sequence heterogeneity among our proteins, the common

**Table 3.** Populations of the four states shown in *Figure 3* resulting from the two independent metadynamics metainference simulations performed for each of the six LCs.

The population of the H state, which is supposed to be a fingerprint specific for AL-LCs, is in bold.

| % | H3 | H7 | H18 | AL55 | M7 | M10 |
|---|---|---|---|---|---|---|
| $L_B$ | 62.0 ± 0.4 | 72.3 ± 2.5 | 22.9 ± 2.8 | 48.3 ± 0.1 | 46.8 ± 0.1 | 48.4 ± 0.3 |
| $L_S$ | 33.0 ± 0.2 | 15.2 ± 2.7 | 38.4 ± 2.1 | 32.5 ± 2.0 | 35.0 ± 0.1 | 49.1 ± 1.5 |
| G | 0.2 ± 0.1 | 0.8 ± 0.4 | 33.8 ± 3.8 | 10.5 ± 1.4 | 17.6 ± 0.1 | 1.8 ± 0.6 |
| H | **4.8 ± 0.5** | **11.7 ± 0.5** | **5.0 ± 1.0** | **8.7 ± 0.5** | **0.6 ± 0.2** | **0.8 ± 0.6** |

AL, amyloidosis; LC, light chain.

peptide analyses were not performed. Therefore, the relative deuterium exchange at different HDX time points of 0.5–240 min with respect to zero exchange time was used to compare the dynamics of individual proteins. The average uptake at different time points for selected regions including residues 34–50 and 152–180 is included in *Figure 5—figure supplement 5*. The individual peptide mapping at all HDX time points is given in *Figure 5—figure supplement 6*. The deuterium uptakes at 30 min HDX time showed the most pronounced differences between different proteins, which were chosen to illustrate the key structural features in the main figure panel (*Figure 5* and *Figure 5—videos 1–4*).

HDX-MS analysis revealed subtle structural dynamics of the individual proteins. The most significant difference between the AL and MM-LCs is observed for residues 34–50, which are part of both the VL-VL dimerization interface and, more importantly in the context of this work, the CL-VL interface. These residues show significantly higher deuterium uptakes in all H-proteins, with H3 being the highest, implying that AL-LCs dimeric interfaces (VL-VL and CL-VL) are more dynamic and hence significantly more destabilized than in M10 (*Figure 5* and *Figure 5—figure supplement 5*). The highly dynamic VL-VL interface of H3 also correlates well with its open VL-VL interface in a crystal structure (PDB: 8P89), which houses two nanobodies interacting with each VL in a dimeric structure (*Broggini et al., 2023*). On the other hand, residues 54–70, which are not part of either interface, show a higher deuterium uptake and hence more dynamics in M10 protein, which may be a result of redistribution of dynamics in the regions away from the rigid VL-VL interface to stabilize the overall structure as also observed for other proteins previously (*Puri and Hsu, 2022*; *Ko et al., 2019*; *Figure 5* and *Figure 5—figure supplement 5*). In contrast, the VL-CL hinge regions (residues 110–120) show homogeneously high flexibility in all the proteins except H7, indicating their higher accessible surface areas (*Figure 5*). As expected, the CL domains from residues 170–200 also show a similar pattern of average deuterium uptakes and hence flexibility in AL- and MM-LCs, with a minor difference contributed by the rigid VL-CL interface containing residues 165–180 (*Figure 5—figure supplements 1 and 4B*). This region shows significantly less deuterium uptake (rigid) in both AL and MM proteins compared with other peptides in the CL domain (*Figure 5—figure supplements 1–4B*). However, comparing the average uptake for this region (residue 152–180) between AL and MM proteins shows that H3 and AL55 have higher uptake than M10. In contrast, H7 is an exception with the lowest deuterium uptake in this region (*Figure 5—figure supplement 5, panel 152–180*). These data are particularly interesting in the light of our simulations. The dimeric conformations identified in the H state (*Figure 3*) are characterized by higher accessibility for the CL-VL interface, which is in agreement with the increased accessibility for the regions 34–50 on the VL and 159–180 in the CL observed in the HDX-MS analysis. Notably, the H state of H7 is the only one in which the CL-CL interface is remarkably compact (see *Figure 4—figure supplement 2*), consistent with the lower HDX for residues 152–180 observed in H7. Overall, the HDX-MS data provide an independent validation of the H state predicted from our conformational ensembles.

## Discussion

Understanding the molecular determinants of AL has been hampered by its high-sequence variability in contrast to its highly conserved 3D structure (*Absmeier et al., 2023*). In this work, building on our previous studies highlighting susceptibility to proteolysis as a property that can discriminate

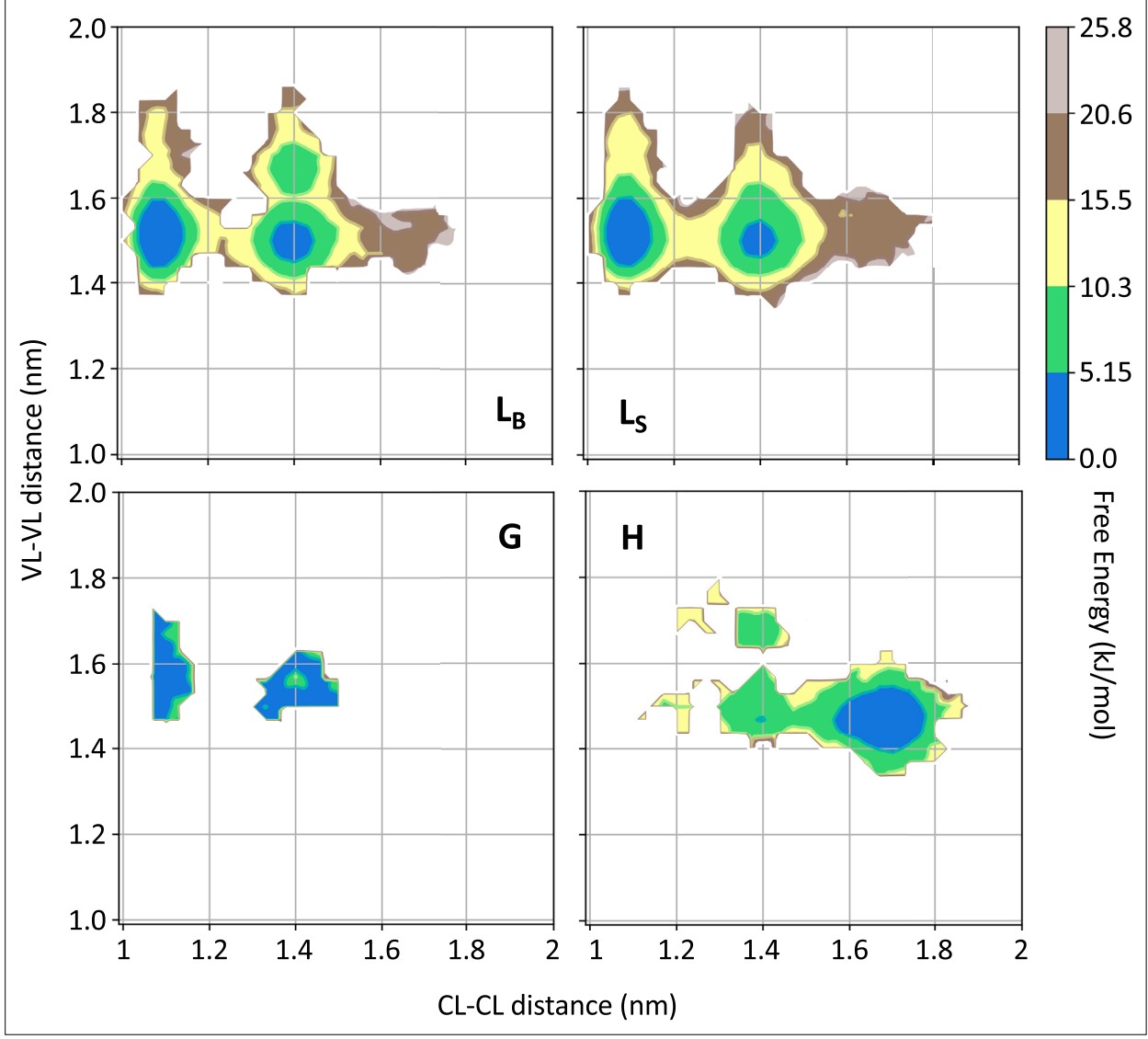

**Figure 4.** Free energy surfaces for the four substates identified in *Figure 3* in the case of the first H3 metainference simulation. The x-axis shows the distance between the centers of mass of the constant domains, while the y-axis shows the distance between the centers of mass of the variable domains. The free energy is shown with color and isolines every $2k_BT$ corresponding to 5.16 kJ/mol. See also *Figure 4—figure supplements 1–6* for the same analysis on the other simulations.

The online version of this article includes the following figure supplement(s) for figure 4:

**Figure supplement 1.** Free energy surfaces for the four substates identified in *Figure 3* in the case of the second H3 metainference simulation.

**Figure supplement 2.** Free energy surfaces for the four substates identified in *Figure 3* in the case of the first (top) and second (bottom) H7 metainference simulation.

**Figure supplement 3.** Free energy surfaces for the four substates identified in *Figure 3* in the case of the first (top) and second (bottom) H18 metainference simulation.

**Figure supplement 4.** Free energy surfaces for the four substates identified in *Figure 3* in the case of the first (top) and second (bottom) AL55 metainference simulation.

**Figure supplement 5.** Free energy surfaces for the four substates identified in *Figure 3* in the case of the first (top) and second (bottom) M7 metainference simulation.

**Figure supplement 6.** Free energy surfaces for the four substates identified in *Figure 3* in the case of the first (top) and second (bottom) M10 metainference simulation.

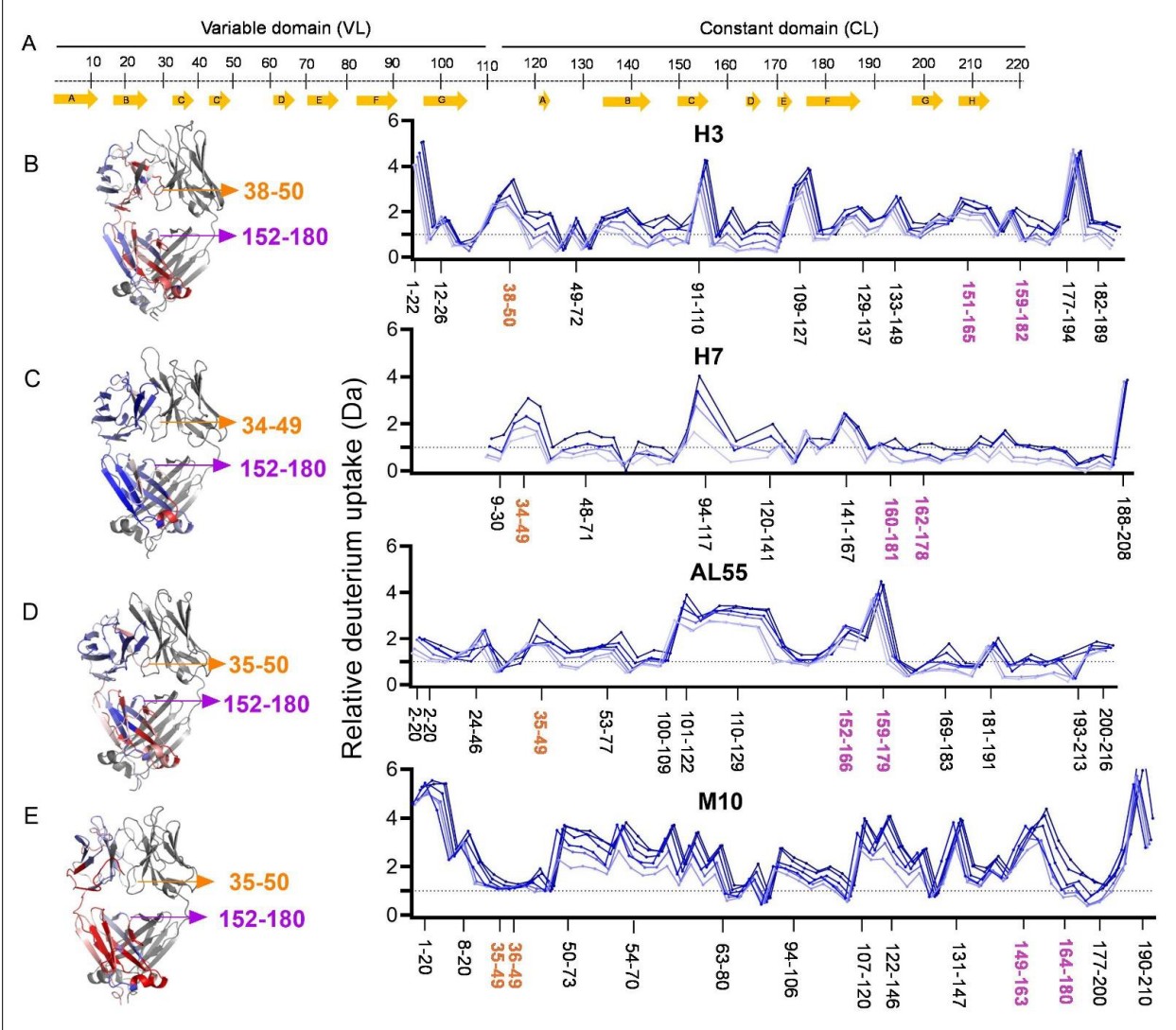

**Figure 5.** Hydrogen-deuterium exchange mass spectrometry analysis. (**A**) The top panel represents the simplified presentation of the primary structure of a light chain, including variable domain (VL) and constant domain (CL). The location of β-strands according to *Chothia and Lesk* (*Al-Lazikani et al., 1997*). (**B**) The structural mapping and butterfly plot of relative deuterium uptake (Da) of H3. Chain A of H3 structure is colored in a gradient of blue–white–red for an uptake of 0–30% at an exchange time of 30 min. Chain B is shown in gray (right-hand panel). The butterfly plot showing relative deuterium uptake at all time points from 0.5 to 240 min on a gradient of light to dark blue (left-hand panel). (**C–E**) are the figures corresponding to H7, AL55, and M10, respectively, with the same color coding as in (**B**). The overall sequence coverage for all proteins was >90%, with a redundancy of >4.0. The VL-VL domains interface peptides covering amino acid residues 34–50 are labeled in orange, while the CL-CL interface region containing 152–180 amino acids is labeled in magenta. Collectively, they form VL-CL interface, which is important to define the H state.

The online version of this article includes the following video and figure supplement(s) for figure 5:

**Figure supplement 1.** H3 HDX coverage and uptake.

**Figure supplement 2.** H7 HDX coverage and uptake.

**Figure supplement 3.** AL55 HDX coverage and uptake.

**Figure supplement 4.** M10 HDX coverage and uptake.

**Figure supplement 5.** Hydrogen-deuterium exchange kinetics of deuterium uptake at each time points from 0 to 240 min for the selected peptides.

**Figure supplement 6.** Structural mapping of relative deuterium uptake on chain A of each dimeric light chain: relative deuterium uptake of H3, H7, AL55, and M10 at different time points of 0.5–240 min on a scale of 0–30% uptake.

**Figure 5—video 1.** 3D structure of H3, color-coded by hydrogen-deuterium exchange and rotating 360°.
https://elifesciences.org/articles/102002/figures#fig5video1

**Figure 5—video 2.** 3D structure of H7, color-coded by hydrogen-deuterium exchange and rotating 360°.

*Figure 5 continued on next page*

*Figure 5 continued*

https://elifesciences.org/articles/102002/figures#fig5video2

**Figure 5—video 3.** 3D structure of AL55, color-coded by hydrogen-deuterium exchange and rotating 360°.

https://elifesciences.org/articles/102002/figures#fig5video3

**Figure 5—video 4.** 3D structure of M10, color-coded by hydrogen-deuterium exchange and rotating 360°.

https://elifesciences.org/articles/102002/figures#fig5video4

between AL-LC and MM-LC, as well as the role of conformational dynamics in protein aggregation, we characterized LC conformational dynamics under the assumption that AL-LC proteins, despite their sequence diversity, may share a property that emerges at the level of their dynamics. We combined SAXS measurements with MD simulations under the integrative framework of metainference to generate conformational ensembles representing the native state conformational dynamics of four AL-LC and two MM-LC. While SAXS alone already indicated possible differences, its combination with MD allowed us to observe a possible low-populated state, which we refer to as state H, characterized by well-separated VL and CL dimers and a perturbed CL-CL interface, which is significantly populated in AL-LCs while only marginally populated in MM-LCs. Notably, high-energy states associated with amyloidogenic proteins have been previously identified in the case of SH3 (*Neudecker et al., 2012*) and β2m (*Le Marchand et al., 2018*). Here, HDX measurements allowed us to independently validate this state by observing increased accessibility in CL-VL interface regions. Importantly, given the limited size of our LC set, we observed in *Peterle et al., 2021* that a comparison of $\lambda$-6 LCs by HDX-MS revealed differences between AL and MM LCs in a region overlapping with the one reported here (residues 35–50). Specifically, this region exhibited increased deuterium uptake in AL compared with MM proteins. A T32N substitution was shown to mitigate aggregation propensity and reduce deuterium uptake in the AL protein, while the reverse substitution, N32T, applied to the germline sequence, increased it (*Peterle et al., 2021*). Finally, the H state observed here is comparable to that previously observed for an aggregation-prone mutation in a monomeric κ LC (*Weber et al., 2018*).

Having established a conformational fingerprint for AL-LC proteins, it would be tempting to identify possible mutations that could be associated with the presence of the H state. Comparing the sequences and structures of M7 and H18, both of which belong to the *IGLV3-19*01* germline, we can identify a single mutation, A40G, that could easily be associated with the appearance of the H state in H18. This mutation is located in the 37–43 loop, which H/D exchange showed to be more accessible in our three AL-LCs than in our MM-LC (see *Figure 5* and *Figure 5—figure supplement 5*), and it breaks a hydrophobic interaction with the methyl group of T165, as observed in the crystal structure of M7 (PDB: 5MVG and *Figure 5—figure supplement 6*), potentially making T165 more accessible in H18 than in M7 (see *Figure 5* and *Figure 5—figure supplement 5*). Comparing the H18 and M7 sequences with the germline reference sequence, we see that position 40 in *IGLV3-19*01* is a glycine (see *Figure 6—figure supplement 1*). This would suggest the intriguing interpretation that the G40A mutation in M7 may increase the interdomain stability compared with the germline sequence, making it less susceptible to aggregation. However, it should also be noted that, while this framework position is a glycine in H3, H7, and AL55, it is also a glycine in M10. Previous research has often focused on identifying, on a case-by-case basis, the key mutations that may be considered responsible for the emergence of the aggregation propensity under the assumption that such aggregation propensity should not be present in germline sequences, but this assumption may be misleading given the observation that few germlines are strongly over-represented in AL, suggesting that these starting germline sequences may be inherently more aggregation-prone than the germline genes that are absent or rarely found in AL patients. More generally, by comparing our AL-LC sequences with their germline references (*Figure 6—figure supplements 2–7*), we observe that all mutations fall exclusively in the variable domain, allowing us to exclude for these systems a direct role for residues in the linker region, as observed in *Weber et al., 2018* and *Nokwe et al., 2015*, or in the constant domain, as observed in *Rottenaicher et al., 2023*. Many mutations fall in the CDR regions, as expected, but others are found in the framework regions, both near the dimerization interface and in other regions of the protein. Somatic mutations in the dimerization interfaces have been identified previously as possibly responsible for toxicity of AL-LC (*Garofalo et al., 2021*). Regarding mutations in the CDRs, it has been suggested that AL-LC proteins may exhibit frustrated CDR2 and CDR3 loops, with

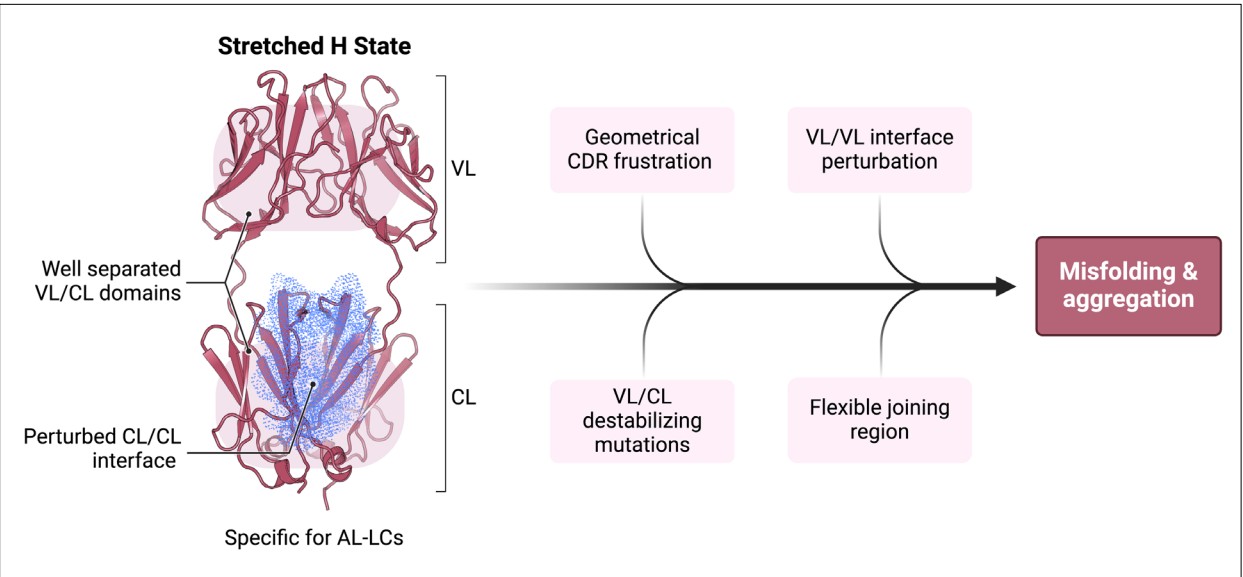

**Figure 6.** Schematic representation summarizing our findings in the context of previous work on the biophysical properties of amyloidogenic light chains (LCs). We propose that the H state is the conformational fingerprint distinguishing amyloidosis (AL) LCs from other LCs, which, together with other features, contributes to the amyloidogenicity of AL LCs.

The online version of this article includes the following figure supplement(s) for figure 6:

**Figure supplement 1.** Zoom-in on the crystal structure of M7 to show the hydrophobic contact between A40 in the variable domain (VL) and T165 in the constant domain (CL).

**Figure supplement 2.** Pairwise sequence alignment between H3 and its corresponding germline as identified by igBLAST using the IGMT databases.

**Figure supplement 3.** Pairwise sequence alignment between H7 and its corresponding germline as identified by igBLAST using the IGMT databases.

**Figure supplement 4.** Pairwise sequence alignment between H18 and its corresponding germline as identified by igBLAST using the IGMT databases.

**Figure supplement 5.** Pairwise sequence alignment between AL55 and its corresponding germline as identified by igBLAST using the IGMT databases.

**Figure supplement 6.** Pairwise sequence alignment between M7 and its corresponding germline as identified by igBLAST using the IGMT databases.

**Figure supplement 7.** Pairwise sequence alignment between M10 and its corresponding germline as identified by igBLAST using the IGMT databases.

few key residues populating the left-hand alpha helix or other high-energy conformations (*Pradhan et al., 2023*), resulting in the destabilization of the VL. In *Figure 6—figure supplements 2–7*, we have analyzed the Ramachandran plot obtained from our conformational ensembles, focusing only on those residues that most populate the left-hand alpha helix region, which are marked with a red circle symbol and whose Ramachandran is reported. Our data indicate the presence of residues populating left-hand alpha regions in the Ramachandran plot, but these are also found in the case of MM-LCs, so our simulations do not allow us to confirm or exclude this mechanism in our set of protein systems.

In conclusion, our study provides a novel, complementary, perspective on the determinants of the misfolding propensity of AL-LCs that we schematize in *Figure 6*. The identification of a high-energy state, with perturbed CL dimerization interfaces, extended linkers, and accessible regions in both the VL-CL and VL-VL interfaces, may be the common feature interplaying with specific properties shown by previous work, including the direct or indirect destabilization of both the VL-VL and CL-CL dimerization interfaces (*Rottenaicher et al., 2021*; *Rennella et al., 2019*; *Rottenaicher et al., 2023*; *Garofalo et al., 2021*; *Pradhan et al., 2023*; *Peterson et al., 2010*; *Kazman et al., 2020*). Our conformational fingerprint is also consistent with the observation that protein stability does not fully correlate with the tendency to aggregate, whereas susceptibility to proteolysis and conformational dynamics may better capture the differences between AL-LC and MM-LC. In this context, our data allow us to rationally

suggest that targeting the constant domain region at the CL-VL interface, which is more labile in the H state, may be a novel strategy to search for molecules against LC aggregation in AL.

# Materials and methods

## Key resources table

| Reagent type (species) or resource | Designation | Source or reference | Identifiers | Additional information |
|---|---|---|---|---|
| Strain, strain background (*Escherichia coli*) | DH5α | NEB 5-alpha | Cat# C2987H | Chemical competent cells |
| Strain, strain background (*E. coli*) | BL21(DE3) | NEB | Cat# C2527H | Chemical competent cells |
| Peptide, recombinant protein | H3 | *Oberti et al., 2017* | NA | |
| Peptide, recombinant protein | H7 | *Oberti et al., 2017* | NA | |
| Peptide, recombinant protein | AL55 | *Puri et al., 2025* | NA | |
| Peptide, recombinant protein | H18 | *Oberti et al., 2017* | NA | |
| Peptide, recombinant protein | M7 | *Oberti et al., 2017* | NA | |
| Peptide, recombinant protein | M10 | *Oberti et al., 2017* | NA | |
| Recombinant DNA reagent (plasmid) | pET21b(+)-H3 | *Oberti et al., 2017* | NA | |
| Recombinant DNA reagent (plasmid) | pET21b(+)-H7 | *Oberti et al., 2017* | NA | |
| Recombinant DNA reagent (plasmid) | pET21b(+)-AL55 | *Puri et al., 2025* | NA | |
| Recombinant DNA reagent (plasmid) | pET21b(+)-H18 | *Oberti et al., 2017* | NA | |
| Recombinant DNA reagent (plasmid) | pET21b(+)-M7 | *Oberti et al., 2017* | NA | |
| Recombinant DNA reagent (plasmid) | pET21b(+)-M10 | *Oberti et al., 2017* | NA | |
| Software, algorithm | PLGS | Waters | RRID:SCR_016664 | Section 'Hydrogen-deuterium mass exchange spectrometry' http://www.waters.com/ |
| Software, algorithm | DynamX | Waters | | Section 'Hydrogen-deuterium mass exchange spectrometry' http://www.waters.com/ |
| Software, algorithm | PyMOL | Schrodinger LLC | RRID:SCR_000305 | Section 'Hydrogen-deuterium mass exchange spectrometry' http://www.pymol.org/ |
| Software, algorithm | GraphPad Prism | GraphPad | RRID:SCR_002798 | Section 'Hydrogen-deuterium mass exchange spectrometry' http://www.graphpad.com/ |
| Software, algorithm | ATSAS 3 package | *Manalastas-Cantos et al., 2021* | RRID:SCR_015648 | Section 'Small-angle X-ray scattering' http://www.embl-hamburg.de/biosaxs/atsas-online/ |
| Software, algorithm | GROMACS 2019 | *Abraham et al., 2015* | RRID:SCR_014565 | Section 'Molecular dynamics simulations' https://www.gromacs.org/ |

*Continued on next page*

*Continued*

| Reagent type (species) or resource | Designation | Source or reference | Identifiers | Additional information |
|---|---|---|---|---|
| Software, algorithm | PLUMED2 v2.9 | *Tribello et al., 2014* | RRID:SCR_021952 | Section 'Molecular dynamics simulations' https://www.plumed.org/ |
| Other | Hi prep QFF- 20 ml column | Cytiva | Product code 28936543 | Section 'Protein production and purification' |
| Other | Superdex 200 10/300 increase | Cytiva | Cat# 28990946 | Section 'Protein production and purification' |
| Other | Superdex 75 10/300 GL | Cytiva | Cat# 17517401 | Section 'Protein production and purification' |
| Other | Immobilized pepsin digestion column | Waters | (Waters Enzymate BEH Pepsin, 2.1 × 30 mm) | Section 'Hydrogen-deuterium mass exchange spectrometry' |
| Other | Isopropyl-β-D-thiogalactopyranoside (IPTG) | Himedia | RM2578 | Section 'Protein production and purification' |
| Other | Guanidinium chloride | Sigma-Aldrich | G3272-10KG | Section 'Protein production and purification' |
| Other | $D_2O$ | Sigma-Aldrich | CAS 7789-20-0 | Section 'Hydrogen-deuterium mass exchange spectrometry' |

## LC production and purification

Recombinant AL- (H3, H7, H18, and AL5) and M- (M7, M10) proteins were produced and purified from the host *Escherichia coli* strain BL21(DE3). First, the competent BL21(DE3) cells were transformed with plasmid pET21(b+), which contains genes encoding H3, H7, H18, AL55, M7, and M10 proteins. The transformed cells were selected for each plasmid by growing them on LB agar plates containing the antibiotic ampicillin at a final concentration of 100 μg/ml. For overexpression of protein, one colony was picked from each plate and grown overnight in 20 ml of LB broth containing ampicillin at a final concentration of 100 μg/ml. The overnight-grown cells were then used to inoculate a secondary culture in 1 l of LB broth. The cells were grown until the turbidity ($OD_{600nm}$) reached between 0.6 and 0.8 and protein expression was subsequently induced by adding 0.5 mM isopropyl-β-D-thiogalactopyranoside for 4 h. The bacterial cells containing overexpressed LCs were then harvested using a Backman Coulter centrifuge at 6000 rpm for 20 min at 4°C. All the proteins were overexpressed as inclusion bodies. For protein purification, the inclusion bodies were isolated by cell lysis induced by sonication. The purification of inclusion bodies was performed by washing them with buffer containing 10 mM Tris (pH 8) and 1% triton X 100. The purified inclusion bodies were unfolded with buffer containing 6.0M guanidinium chloride (GdnHCl) for 4 h at 4°C. The unfolded LCs were then refolded in a buffer containing reduced and oxidized glutathione to assist in disulfide bond formation. The refolded proteins were subjected to anion exchange and SEC steps for final purification. The level of protein purity was checked on 12% sodium dodecyl sulfate-polyacrylamide gel electrophoresis gels. The final protein concentration was measured using molecular weight and extinction coefficient of individual proteins. The purified proteins were stored at –20°C for further use. Size exclusion coupled multiangle light scattering confirmed the dimeric assembly of purified proteins (*Figure 1—figure supplement 2*), which were used for SAXS and HDX-MS experiments mentioned below.

## Small-angle X-ray scattering

For SAXS analysis, H3 was diluted to 3.4 mg/ml, H7 was diluted to 3.4 mg/ml, H18 was diluted to 2.8 mg/ml, AL55 was diluted to 2.6 mg/ml, M7 was diluted to 3.6 mg/ml, in 20 mM Tris-HCl, 150 mM NaCl, pH 8. H3, H7, and M7 batch data were collected at the P12 BioSAXS beamline of the EMBL Hamburg Synchrotron (*Blanchet et al., 2015*), while AL55 batch data and H18 and M10 SEC data were collected at the BM29 BioSAXS beamline of the ESRF, Grenoble (*Pernot et al., 2010*). For SEC-SAXS, H18 and M10 were injected into a superdex 200 increase 10/300 GL column previously equilibrated in 20 mM Tris-HCl, 150 mM NaCl, pH 8, at a concentration of 2.8 mg/ml and 6.7 mg/ml, respectively (see also *Figure 1—figure supplement 3*). SAXS data were processed using programs

PRIMUS and GNOM within the ATSAS package (*Manalastas-Cantos et al., 2021*). Data are deposited in the SASBDB (*Valentini et al., 2015*) and available with accession codes SASDVL4, SASDVM4, SASDVN4, SASDVK4, SASDVP4, and SASDVQ4.

## Molecular dynamics simulations

The available crystallographic structures of H3, H7, and M7 (PDB: 5mtl, 5muh, and 5mvg, respectively; *Oberti et al., 2017*) were used as starting conformations, using Modeller to add missing residues (*Webb and Sali, 2016*). H18 and AL55 were modeled by homology modeling using SwissModel (*Waterhouse et al., 2018*), while M10 was modeled using AF2 (*Jumper et al., 2021*). Simulations were performed using GROMACS 2019 (*Abraham et al., 2015*) and the PLUMED2 software (*Tribello et al., 2014*; *Bonomi and Camilloni, 2017*; *Bonomi, 2019*), using AMBER-DES force field and TIP4P-D water (*Piana et al., 2020*; *Piana et al., 2015*). During in-vacuum minimization, RMSD (Root mean square deviation)-restraints were imposed to enhance the symmetry between the two constant and the two variable domains. The systems were solvated in a periodic dodecahedron box, initially 1.2 nm larger than the protein in each direction, neutralized with Na and Cl ions to reach a salt concentration of 10 mM, then minimized and equilibrated at the temperature of 310 K and pressure of 1 atm using the Berendsen thermostat and barostat. Two independent 900-ns-long plain MD simulations were run to generate reliable and independent starting conformations for the M&M simulations (*Bonomi et al., 2016b*; *Bonomi et al., 2016a*; *Löhr et al., 2017*). Also, 30 conformations were extracted from each simulation and duplicated by inverting the two chains to obtain 60 starting conformations symmetrically distributed with respect to chain inversion.

M&M production simulations were run in duplicate using 60 replicas, each replica evolved for ~1 µs (*Table 2*). Simulations were performed in the NPT ensemble maintaining the temperature at 310 K with the Bussi thermostat (*Bussi et al., 2007*) and the pressure of 1 atm with the Parrinello–Rahman barostat *Parrinello and Rahman, 1981*; the electrostatic was treated using the particle mesh Ewald scheme with a short-range cut-off of 0.9 nm, and van der Waals interaction cut-off was set to 0.9 nm. To reduce the computational cost, the hydrogen mass repartitioning scheme was used (*Hopkins et al., 2015*): the mass of heavy atoms was repartitioned into the bonded hydrogen atoms using the heavyh flag in the *pdb2gmx* tool, and the LINCS algorithm was used to constraint all bonds, allowing to use a time step of 5 fs. In these simulations, Parallel Bias Metadynamics (*Pfaendtner and Bonomi, 2015*) was used to enhance the sampling, combined with well-tempered metadynamics and the multiple-walker scheme, where Gaussians with an initial height of 1.0 kJ/mol were deposited every 0.5 ps using a bias factor of 10. Five CVs(Collective Variable) were biased, including combinations of phi/psi dihedral angles of the linker regions (i.e., residues connecting variable and constant domains) in the two chains, combinations of chi dihedral angles of the linker regions in the two chains, combination of inter-domain contacts between the variable and the constant domains. The width of the Gaussians was 0.07, 0.12, and 120 for the combination of phi/psi, of chi dihedral angles and combination of contacts, respectively. Metainference was used to include SAXS restraints, using the hySAXS hybrid approach described in *Paissoni et al., 2020*, *Paissoni and Camilloni, 2021*, and *Ballabio et al., 2023*. A set of 13 representative SAXS intensities at different scattering angles, ranging between 0.015 Å$^{-1}$ and 0.25 Å$^{-1}$ and equally spaced, was used as restraints. These intensities were extracted from experimental data after performing regularization with the Distance Distribution tool of Primus, based on Gnom (*Manalastas-Cantos et al., 2021*). Metainference was applied every five steps using a single Gaussian noise per data point and sampling a scaling factor between experimental and calculated SAXS intensities with a flat prior between 0.5 and 1.5. The aggregate sampling from the 60 replicas was reweighted using the final metadynamics bias to obtain a conformational ensemble where each conformation has an associated statistical weight (*Branduardi et al., 2012*). Convergence and error estimates were assessed by the inspection of the two replicated M&M run. SAXS data were then recalculated using crysol (*Manalastas-Cantos et al., 2021*). All relevant simulation data are available on Zenodo (cf. Dataset S1 in *Supplementary file 1*).

## Hydrogen-deuterium mass exchange spectrometry

SYNAPT G2-HDMS system (Waters Corporation, USA) equipped with a LEAP robotic liquid handler was used to perform HDX-MS measurements in a fully automated mode as described previously (*Puri and Hsu, 2022*; *Ko et al., 2019*; *Masson et al., 2019*; *Puri et al., 2022*). The data collection was

carried out by a 20-fold dilution of H3, H7, AL55, and M10 proteins (100 μM) with the labeling buffer 1× phosphate-buffered saline prepared in $D_2O$ (pD 7.4) to trigger HDX for 0, 0.5, 1,10, 30, 120, and 240 min at 25°C in technical triplicates. Each reaction was quenched by mixing the labeled protein with quench buffer (50 mM sodium phosphate, 250 mM TCEP(tris(2-carboxyethyl)phosphine), 3.0 M GdnHCl [pH 2]) in a 1:1 ratio at 0°C. Online digestion was then performed using an immobilized pepsin digestion column (Waters Enzymate BEH Pepsin, 2.1 × 30 mm). The digested peptides were trapped using a C18 trapping column (Acquity BEH VanGuard 1.7 μm, 2.1 × 5.0 mm) and separated by a linear acetonitrile gradient of 5–40%. Protein Lynx Global Server (PLGS) and DynamX (Waters Corporation) were used to identify the individual peptides, and subsequently, data processing using parameters: maximum peptide length of 25, the minimum intensity of 1000; minimum ion per amino acid of 0.1; maximum MH+ error of 5 ppm; and a file threshold of 3. A reference molecule [(Glu1)-fibrinopeptide B human (CAS# 103213-49-6, Merck, USA)] was used to lock mass with an expected molecular weight of 785.8426 Da. The obtained peptide coverage of H3, H7, AL55, and M10 was 98.6, 92.5, 98.6, and 99.1%, respectively, with a redundancy of >4.0. Backbone amide groups exhibited a relative deuterium uptake (with respect to the zero exchange time data) of 0–30% within 4 h of exchange time. The relative deuterium uptake data at different HDX times were then used to generate heat maps for each amino acid. The obtained data were mapped on individual protein structures in a gradient of blue–white–red showing 0–30% of uptake, respectively (*DeLano, 2025*). Red denotes the dynamic peptides, while peptide colors in blue are rigid. The common peptide analysis between different model LCs was not performed in this analysis as they all generate different peptides due to both sequence heterogeneity in LCs, especially in the VL domain, and non-specific cleavage of pepsin enzyme used to generate peptides after deuterium exchanged for MS analysis. Therefore, all interpretation was done on individual relative exchange data.

## Acknowledgements

We acknowledge Martin A Schroer and Dmitri Svergun at EMBL Hamburg, and Sonia Longhi at AFMB Marseille, for discussion and support on the SAXS data acquisition and analysis. We acknowledge PRACE for awarding us access to Piz Daint at CSCS, Switzerland. We acknowledge CINECA for an award under the ISCRA initiative for the availability of high-performance computing resources and support. SP acknowledges Fondazione Veronesi for a postdoctoral fellowship. We also acknowledge Dr. Min-Feng Karen Hsu and Mr. Yong-Sheng Wang for the protein quality controls, and Dr. Shu-Yu Lin and Mr. Ming-Jie Tsai of the Academia Sinica Common Mass Spectrometry Facilities (AS-CFII-111-209), funded by the Academia Sinica Core Facility and Innovative Instrument Project, for supporting the HDX-MS experiments.

## Additional information

### Funding

| Funder | Grant reference number | Author |
|---|---|---|
| Ministero dell'Università e della Ricerca | PRIN 20207XLJB2 | Stefano Ricagno |
| Fondazione Cariplo | Telethon GJC23044 | Stefano Ricagno |
| Ministero della Salute | #GR-2018-12368387 | Stefano Ricagno |
| Fondazione AIRC per la ricerca sul cancro ETS | IG 2024 ID 30307 | Stefano Ricagno |
| Academia Sinica | AS-CDA-109- L08 | Shang-Te Danny Hsu |
| National Science and Technology Council | 113-2123-M-001-010- | Shang-Te Danny Hsu |
| National Science and Technology Council | 110-2113-M-001-050-MY3 | Shang-Te Danny Hsu |

| Funder | Grant reference number | Author |
|---|---|---|
| National Science and Technology Council | 113-2811-M-001-110 | Manoj K Sriramoju |
| Academia Sinica | AS-IV-114-L04 | Shang-Te Danny Hsu |

The funders had no role in study design, data collection and interpretation, or the decision to submit the work for publication.

### Author contributions

Cristina Paissoni, Formal analysis, Investigation, Visualization, Methodology; Sarita Puri, Formal analysis, Investigation, Visualization, Methodology, Writing – original draft, Writing – review and editing; Luca Broggini, Manoj K Sriramoju, Rosaria Russo, Valentina Speranzini, Investigation; Martina Maritan, Investigation, Visualization; Federico Ballabio, Data curation; Mario Nuvolone, Giampaolo Merlini, Giovanni Palladini, Shang-Te Danny Hsu, Supervision, Writing – review and editing; Stefano Ricagno, Conceptualization, Supervision, Funding acquisition, Methodology, Writing – original draft, Project administration, Writing – review and editing; Carlo Camilloni, Conceptualization, Supervision, Methodology, Writing – original draft, Project administration, Writing – review and editing

### Author ORCIDs

Rosaria Russo ![ORCID] https://orcid.org/0000-0003-0011-5775
Federico Ballabio ![ORCID] https://orcid.org/0000-0001-5702-3674
Stefano Ricagno ![ORCID] https://orcid.org/0000-0001-6678-5873
Carlo Camilloni ![ORCID] https://orcid.org/0000-0002-9923-8590

Reviewer #1 (Public review): https://doi.org/10.7554/eLife.102002.3.sa1
Reviewer #2 (Public review): https://doi.org/10.7554/eLife.102002.3.sa2
Reviewer #3 (Public review): https://doi.org/10.7554/eLife.102002.3.sa3
Author response https://doi.org/10.7554/eLife.102002.3.sa4

## Additional files

### Supplementary files

Supplementary file 1. Supplementary tables. (a) Pairwise sequence identity (above diagonal) and similarity (below diagonal) for the six systems under study. On the diagonal is reported the germline identified by igBLAST using the IGMT database. (b) HDX-MS data summary.

MDAR checklist

### Data availability

Small-angle x-ray scattering data have been deposited in SASBDB under accession codes SASDVL4, SASDVM4, SASDVN4, SASDVK4, SASDVP4, and SASDVQ4. All molecular dynamics simulation trajectories and associated statistical weights are available on Zenodo.

The following datasets were generated:

| Author(s) | Year | Dataset title | Dataset URL | Database and Identifier |
|---|---|---|---|---|
| Ballabio F | 2024 | Immunoglobulin light chain H3 | https://www.sasbdb.org/data/SASDVL4 | SASBDB, SASDVL4 |
| Ballabio F | 2024 | Immunoglobulin light chain H7 | https://www.sasbdb.org/data/SASDVM4 | SASBDB, SASDVM4 |
| Ballabio F | 2024 | Immunoglobulin light chain H18 | https://www.sasbdb.org/data/SASDVN4 | SASBDB, SASDVN4 |
| Ballabio F | 2024 | Immunoglobulin light chain AL55 | https://www.sasbdb.org/data/SASDVK4 | SASBDB, SASDVK4 |

*Continued on next page*

*Continued*

| Author(s) | Year | Dataset title | Dataset URL | Database and Identifier |
|-----------|------|---------------|-------------|-------------------------|
| Ballabio F | 2024 | Immunoglobulin light chain M7 | https://www.sasbdb.org/data/SASDVP4 | SASBDB, SASDVP4 |
| Ballabio F | 2024 | Immunoglobulin light chain M10 | https://www.sasbdb.org/data/SASDVQ4 | SASBDB, SASDVQ4 |
| Camilloni C, Paissoni C | 2024 | Simulations data for the paper "A conformational fingerprint for amyloidogenic light chains" | https://doi.org/10.5281/zenodo.12731283 | Zenodo, 10.5281/zenodo.12731283 |

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
