## [Editor Report · eLife Assessment]

This study addresses an **important** and long-standing question regarding the molecular mechanism of protein misfolding in Ig light-chain (LC) amyloidosis (AL), a life-threatening condition. By combining advanced techniques, including small-angle X-ray scattering, molecular dynamics simulations, and hydrogen-deuterium exchange mass spectrometry, the authors provide **convincing** evidence that the ‘H state’ distinguishes amyloidogenic from non-amyloidogenic LCs. These findings not only offer novel insights into LC structural dynamics but also hold promise for guiding therapeutic strategies in AL and will be of particular interest to structural biologists, biophysicists, and many others working on amyloid diseases.

---

## [Referee Report · Reviewer #1 (Public review)]

The study investigates light chains (LCs) using three distinct approaches, with a focus on identifying a conformational fingerprint to differentiate amyloidogenic light chains from multiple myeloma light chains. The study's major contribution is the identification of a low-populated "H state," which the authors propose as a unique marker for AL-LCs. While this finding is promising, the review highlights several strengths and weaknesses. Strengths include the valuable contribution of identifying the H state and the use of multiple approaches, which provide a comprehensive understanding of LC structural dynamics. Weaknesses include a lack of physical insights explaining the changes.

---

## [Referee Report · Reviewer #2 (Public review)]

Summary:

This well-written manuscript addresses an important but recalcitrant problem - molecular mechanism of protein misfolding in Ig light chain (LC) amyloidosis (AL), a major life-threatening form of systemic human amyloidosis. The authors use expertly recorded and analyzed small-angle X-ray scattering (SAXS) data as a restraint for molecular dynamics simulations (called M&M). Six patient-based LC proteins are explored, including four AL and two non-AL. The authors report a partially populated "H-state" determined computationally, wherein the two domains in an LC molecule acquire a straight rather than bent conformation, with an extended interdomain linker; this H-state distinguishes AL from non-AL LCs. H-D exchange mass spectrometry is used to support this conclusion. This is a novel and interesting finding with potentially important translational implications.

Strengths:

Expertly recorded and analyzed SAXS data combined with clever M&M simulations lead to a novel and interesting conclusion, which is supported by limited H-D exchange data.

Stabilization of the CL-CL interface is a good idea that may help protect a subset of AL LCs from misfolding in amyloid.

Computational M&M evidence is convincing and is supported by SAXS data, which are used as restraints for simulations. Although Kratky plots reported in the main MS Fig. 1 show significant differences between the data and the structural model for only one AL protein, AL-55, H-state is also inferred for other AL proteins.

Apparent limitations:

HDX MS results show that residues 35-50 from VL-VL and VL-CL dimerization interface are less protected in AL vs. non-AL proteins, which is consistent with the H-state. However, the small number of proteins yielding useful HDX data (three AL and one non-AL) suggests that this conclusion should be treated with caution. It is unclear whether the conformational heterogeneity depicted in M&M simulations is consistent with HDX results, and whether prior HDX studies of AL and MM LCs are consistent with the conclusions that a particular domain-domain interface is weakened in AL vs. non-AL LCs. The butterfly plots in Fig. 5 could benefit from the X-axis labeling with the peptide fragments.

---

## [Referee Report · Reviewer #3 (Public review)]

Summary:

This study identifies confirmational fingerprints of amylodogenic light chains, that set them apart from the non-amylodogenic ones.

Strengths:

The research employs a comprehensive combination of structural and dynamic analysis techniques, providing evidence that conformational dynamics at VL-CL interface and structural expansion are distinguished features of amylodogenic LCs.

Weaknesses:

The sample size is limited, which may affect the generalizability of the findings. Additionally, the study could benefit from deeper analysis of specific mutations driving this unique conformation to further strengthen therapeutic relevance.

Furthermore. p-value (statistical significance) of Rg difference should be computer. Finally, significance of mutations (SHM?) at the interface, such as A40G should be compared with previous observations. (Garofalo et al., 2021)

---

## [Author Response]

The following is the authors’ response to the original reviews.

**eLife Assessment**
This important study identifies the "H-state" as a potential conformational marker distinguishing amyloidogenic from non-amyloidogenic light chains, addressing a critical problem in protein misfolding and amyloidosis. By combining advanced techniques such as small-angle X-ray scattering, molecular dynamics simulations, and H-D exchange mass spectrometry, the authors provide convincing evidence for their novel findings. However, incomplete experimental descriptions, limitations in SAXS data interpretation, and the way HDX MS data is presented aHect the strength and generalizability of the conclusions. Strengthening these aspects would enhance the impact of this work for researchers in amyloidosis and protein misfolding.

We thank eLife editors and reviewers for their constructive feedback. The manuscript has been improved to provide a more complete description of the experiments and to strengthen the interpretation and presentation of all data. Updated Figures (Figure 2 and Figure 5) and a new Table (Table 2) in the main text provide a more complete and clearer comparison of the SAXS data with MD simulations as well as a clearer representation of the HDX MS data. Additional figures have been added in SI. The text has been extended accordingly and complete materials and methods are now included in the main text. Abstract, introduction and discussion have been revised to improve the overall readability of the manuscript.

**Public Reviews:**

**Reviewer #1 (Public review):**
The study investigates light chains (LCs) using three distinct approaches, with a focus on identifying a conformational fingerprint to diHerentiate amyloidogenic light chains from multiple myeloma light chains. The study's major contribution is identifying a low-populated "H state," which the authors propose as a unique marker for AL-LCs. While this finding is promising, the review highlights several strengths and weaknesses. Strengths include the valuable contribution of identifying the H state and using multiple approaches, which provide a comprehensive understanding of LC structural dynamics. However, the study suHers from weaknesses, particularly in interpreting SAXS data, lack of clarity in presentation, and methodological inconsistencies. Critical concerns include high error margins between SAXS profiles and MD fits, unclear validation of oligomeric species in SAXS measurements, and insuHicient quantitative cross-validation between experimental (HDX) and computational data (MD). This reviewer calls for major revisions including clearer definitions, improved methodology, and additional validation, to strengthen the conclusions.

We thank the reviewer for the supportive comments, in the revised version of the manuscript we have focused on improving the clarity and completeness of our work. We are sorry for example to not have made previously clear enough that the comparison of SAXS with MD simulation was not that shown in the main text in Figure 1 and Table 1 (this is the comparison with single structures) but that reported in the SI (previously Figure S1 and Table S2, showing very good fits). These data have been moved in the main text in the reworked Figure 2 and new Table 2. We have also improved the presentation of the HDX MS data in Figure 5 and in the text adding also additional analysis in SI. Materials and methods are now completely moved in the main text. We generally revised the manuscript for clarity.

**Reviewer #2 (Public review):**
Summary:This well-written manuscript addresses an important but recalcitrant problem - the molecular mechanism of protein misfolding in Ig light chain (LC) amyloidosis (AL), a major life-threatening form of systemic human amyloidosis. The authors use expertly recorded and analyzed smallangle X-ray scattering (SAXS) data as a restraint for molecular dynamics simulations (called M&M) and to explore six patient-based LC proteins. The authors report that a highly populated "H-state" determined computationally, wherein the two domains in an LC molecule acquire a straight rather than bent conformation, is what distinguishes AL from non-AL LCs. They then use H-D exchange mass spectrometry to verify this conclusion. If confirmed, this is a novel and interesting finding with potentially important translational implications.

We thank the reviewer for the supportive comments.

Strengths:Expertly recorded and analyzed SAXS data combined with clever M&M simulations lead to a novel and interesting conclusion. Regardless of whether or not the CL-CL domain interface is destabilized in AL LCs explored in this (Figure 6) and other studies, stabilization of this interface is an excellent idea that may help protect at least a subset of AL LCs from misfolding in amyloid. This idea increases the potential impact of this interesting study.

We thank the reviewer for the supportive comments.

Weaknesses:The HDX analysis could be strengthened.

We have extended the analysis and improved the presentation of the HDX data. Figure 5 has been reworked, text has been improved accordingly and additional analysis have been reported in SI.

**Reviewer #3 (Public review):**
Summary:This study identifies conformational fingerprints of amyloidogenic light chains, that set them apart from the non-amyloidogenic ones.

We thank the reviewer for the supportive comments.

Strengths:The research employs a comprehensive combination of structural and dynamic analysis techniques, providing evidence that conformational dynamics at the VL-CL interface and structural expansion are distinguished features of amyloidogenic LCs.

We thank the reviewer for the supportive comments.

Weaknesses:The sample size is limited, which may aHect the generalizability of the findings. Additionally, the study could benefit from deeper analysis of specific mutations driving this unique conformation to further strengthen therapeutic relevance.

We agree, we tried to maximise the size of the sample and this was the best we could do. With respect to the analysis of the mutations, while we tried to discuss some of them also in view of previous works, because our set covers multiple germlines instead than focusing on a single one, this limit our ability to discuss single point mutations systematically, at the same time the discussion of single points mutations has been the focus of many recent works, while our approach provide a diNerent point of view.

**Recommendations for the authors:**

**Reviewer #1 (Recommendations for the authors):**
This study provides an investigation of light chains (LCs) using three distinct approaches, focusing primarily on identifying a conformational fingerprint to distinguish amyloidogenic light chains (AL-LCs) from multiple myeloma light chains (MM-LCs). The authors propose that the presence of a low-populated "H state," characterized by an extended quaternary structure and a perturbed CL-CL interface, is unique to AL-LCs. This finding is validated through hydrogendeuterium exchange mass spectrometry (HDX-MS). The study makes a valuable contribution to understanding the structural dynamics of light chains, particularly with the identification of the H state in AL-LCs. However, significant concerns regarding the interpretation of the SAXS data, clarity in presentation, and methodological rigor must be addressed. I recommend major revisions and resubmission of the work.Major concerns:(1) A critical concern is how the authors ensure that the SAXS profiles represent only dimeric species, given the high propensity of LCs to aggregate. If higher-order aggregates or monomers were present, this would significantly impact the SAXS data and SAXS-MD integration. Some measurements are bulk SAXS, while others are SEC-SAXS, making the study questionable. The authors need to clarify how only dimeric species were measured for the SEC-SAXS analysis, and all assessments of the dimeric state should be shown in the SI. Additionally, complementary techniques such as DLS or SEC-MALS should be used to verify the oligomeric state of the samples. Without this validation, the SAXS profiles may not be reliable.

We added SEC-MALS and SEC-SAXS data in the SI (Figures S20 and S21) as well the SAXS curves shown in log-log plot (Figure S1) that display a flat trend at low q that exclude aggregation. SAXS is very sensitive to oligomers and aggregates and our data do not indicate the presence of those species. When we had indication of possible aggregation in the sample we used SEC-SAXS.

(2) A major problem with the paper is that the claim of the "H state," which is the novelty of the study and serves as a marker of aggregation, is derived from samples where the error between the SAXS profiles and MD fits is extremely high. This casts doubt on whether the structure is indeed resolved by MD. The main conclusion of the paper is derived from weak consistency between experiment and simulation. In AL55, the error between experiment and simulation is greater than 5; for H7, it is higher than 2.8. The residuals show significant error at mid-q values, suggesting that long-range distance correlations (20-10 Å, CL, VL positioning) are not consistent between simulation and experiment. Furthermore, the FES plots of two independent replicas show deviation in the existence of the H state. One shows a minimum in that region, while the other does not. So, how robust is this conclusion? What is the chi-squared value if each replica is used independently? A separate experimental cross-validation is necessary to claim the existence of the H state.

We apologise for the misunderstanding underlying this reviewer comment. The poor agreement mentioned is not between the SAXS and MD simulations, but with the individual structures, and this disagreement led us to perform MD simulations that are in much better agreement with the data (previously Fig. S1 and Table S2). To avoid this misunderstanding, which would indeed weaken our work, we have now moved both the figure and the table in the main text to the updated Figure 2 and the new Table 2.

Regarding the robustness of the sampling, we believe that Table 3 (previously Table 2) clearly shows the statistical convergence of the data, diNerences in the presentation of the free energy are purely interpolation issues. The chi-squares of each replicate are reported in Table 2 (previously Table S2).

(3) There is insuHicient discussion about SAXS computations from MD trajectories. The accuracy of these calculations is crucial to deriving the existing conclusions, and the study's reliance on the PLUMED plugin, which is known to give inaccurate results for SAXS computations, raises concerns. How the solvent is treated in the SAXS computations needs to be explained. Alternative methods like WAXSiS or Crysol should be explored to check whether the SAXS profiles derived from the MD trajectory are consistent across other SAXS computation methods for the major conformers of the proteins.

We have now clarified that while the SAXS calculation to perform Metainference MD were done using PLUMED (that to our knowledge is as accurate as crysol) SAXS curves used for analysis were calculated using crysol.

(4) The HDX and MD results do not seem to correlate well, and there is a disconnect between Figure 2 (SAXS profiles) and Figure 5 (HDX structural interpretation). The authors should quantitatively assess residue-level dynamics by comparing HDX signals with MD-derived HDX signals for each protein. This would provide a cross-validation between the experimental and computational data.

In our opinion our SAXS, MD and HDX MS data provide a consistent picture. Our HDX-MS do not provide per residue data, making a quantitative comparison out of scope. RMSF data do not necessarily need to correlate with the deuterium uptake.

(5) MD simulations are only used to refine the structure of AlphaFold predictions, but the trajectories could help explain why these structures diHer, what stabilizes the dimer, or what leads to the conformational transition of the H state. A lack of analysis regarding the physical mechanism behind these structural changes is a weakness of the study. The authors should dedicate more eHort to analyzing their data and provide physical insights into why these changes are observed.

Our aim was to identify a property that could discriminate between AL and MM LCs. We used MD simulations, not to refine structures, but to explore the conformational dynamics of LCs (starting from either X-ray structures, homology or AlphaFold models), because SAXS data suggested that conformational dynamics could discriminate between AL- and MM-LCs. Simulations allowed us to propose a hypothesis, which we tested by HDX MS. While more insight is always welcome, we believe that we have achieved our goal for now. In the discussion, we present additional analysis of the simulations to connect with previous literature, we agree that more analysis can be done, and also for this reason, all our data are publicly available.

Minor concerns(6) The abstract leans heavily on describing the problem and methods but lacks a clear presentation of key results. Providing a concise summary of the main findings (e.g., the identification of the H state) would better balance the abstract.

We agree with the reviewer and we rewrote the abstract.

(7) In the abstract, the term "experimental structure" is used ambiguously. Since SAXS also provides an experimental structure, it is unclear what the authors are referring to. This should be clarified.

We agree with the reviewer and we rewrote the abstract.

(8) Abbreviations such as VL (variable domain) and CL (constant domain) are not defined, making it harder for readers unfamiliar with the field to follow. Abbreviations should be defined when first mentioned.

We agree with the reviewer and we rewrote the abstract.

(9) The introduction provides a good general context but fails to explicitly define the knowledge gap. Specifically, the structural and dynamic determinants of LC amyloidogenicity are not well established, and this study could be framed as addressing that gap.

We thank the reviewer and we agree this could be better framed, we improved the introduction accordingly.

(10) The introduction does not present the novel discovery of the H state early enough. The unique contribution of identifying this state as a marker for AL-LCs should be mentioned upfront to guide the reader through the significance of the study.

We thank the reviewer and we have now made more explicit what we found.

(11) The therapeutic implications of this research should be highlighted more clearly in the discussion. Examples of how these findings could be utilized in drug design or therapeutic approaches would enhance the study's impact.

We thank the reviewer, but while we think that the H-state could be targeted for drug design, since we do not have data yet we do not want to stress this point more than what we are already doing.

(12) There is an overwhelming use of abbreviations such as H3, H7, H18, M7, and M10 without proper introduction. This makes it diHicult for readers to follow the results, and the average reader may become lost in the details. An introductory figure summarizing the sequences under study, along with a schematic of the dimeric structure defining VL and CL domains, would significantly aid comprehension.

We agree and we tried to better introduce the systems and simplify the language without adding a figure that we think would be redundant.

(13) In Figure 1, add labels to each SAXS curve to indicate which protein they correspond to. Also, what does online SEC-SAXS mean?

Done

(14) The caption of Figure 3 is unclear, particularly with abbreviations like Lb, Ls, G, and H, which are not mentioned in the captions. The authors should define these terms for clarity.

Done

(15) The study claims that the dominant structure of the dimer changes between diHerent LCs. However, Figure 5 shows identical structures for all proteins, raising questions about the consistency between the SAXS and HDX data. This inconsistency is a general problem between the MD and HDX sections, where cross-communication and comparisons are not properly addressed.

We do not claim that the dominant structure of the dimer changes between diNerent LCs, this would also be in contradiction with current literature. We claim a diNerence in a low-populated state. From this point of view using always the same structure is consistent and should simplify the representation of the results. We agree that the manuscript may be not always easy to follow and we thank the reviewer in helping us improving it.

(16) The authors show I(q) vs q and residuals for each protein. The Kratky plots are not suHicient to compare the SAXS computations with the measured profile.

Showing Kratky and residuals is a standard and complementary way to present and compare SAXS data to structures. Chi-square values are also reported. Log-log plots have been added to SI in response to previous comments.

(17) The authors need to explain how they estimate the Rg values (from simulation or SAXS profiles). If they are using simulations, they should compute the Rg values from the simulations for comparison.

Rg values reported in Table 1 are derived from SAXS. Rg from simulations have been added in Table 2.

(18) The evolution of the sampling is unclear. The authors need to show the initial starting conformation in each case and the most likely conformation after M&M in the SI, to demonstrate that their approach indeed caused changes in the initial predictions.

Our approach is not structure refinement and as such the proposed analysis would be misleading. Metainference is meant to generate a statistical ensemble representing the equilibrium conformations that as whole reproduce the data. DiNerences (or not) between initial and selected configurations will not be particularly informative in this context.

(19) The authors should also provide a running average of chi-squared values over time to demonstrate that the conformational ensemble converged toward the SAXS profile.

Our simulations are not driven to improve the agreement with SAXS over time, this is not structure refinement. Metainference is meant to generate a statistical ensemble representing the equilibrium conformations that as whole reproduce the data. The suggested analysis would be a misinterpretation of our simulations. The comparison with SAXS is provided in Figure 2 and Table 2 as mentioned above.

(20) The aggregate simulation time of 120 microseconds is misleading, as each replica was only run for 2-3 microseconds. This should be clarified.

The number reported in the text is accurate and represent the aggregated sampling. The number of replicas for each metainference simulation and their length is reported in Table 2 now moved for clarity from the SI to main text.

(21) It is not clear how the replicas were weighted to compute the SAXS profiles and FES. There are two independent runs in each case, and each run has about 30 replicas. How these replicas are weighted needs to be discussed in the SI.

Done

(22) The methods section is unevenly distributed, with detailed explanations of LC production and purification, while other key methodologies like SAXS+MD integration and HDX are not even mentioned in the main text (they are in the Supporting Information). The authors should provide a brief overview of all methodologies in the main text or move everything to the SI for consistency.

We agree with the reviewer, all methods are now in main text.

**Reviewer #2 (Recommendations for the authors):**
(1) Computational M&M evidence is strong (Figure 3) and is supported by SAXS (used as restraints). However, Kratky plots reported in the main MS Figure 1 show significant diHerences between the data and the structural model only for one protein, AL-55. It is hard for the general reader to see how these SAXS data support a clear diHerence between AL and non-AL proteins. If possible, please strengthen the evidence; if not, soften the conclusions.

We thank the reviewer for the comments. The chi-square (Table 1) and the residuals (Figure 1) are a strong indication of the diNerence. To strengthen the evidence, following also the comment from reviewer 3 we calculated the p-value (<10^-5^) on the significance of the radius of gyration to discriminate AL and MM LCs. We agree that SAXS alone was not enough and this is indeed what prompted us to perform MD simulations.

(2) HDX MS results are cursory and not very convincing as presented. The butterfly plots in Figure 5 are too small to read and are unlabeled so it is unclear which protein is which.

Figure 5 has been reworked for readability. More data have been added in SI.

(3) What labeling time was selected to construct these plots and why?

The deuterium uptakes at 30 min HDX time showed the most pronounced diNerences between diNerent proteins, which were chosen to illustrate the key structural features in the main figure panel (Figure 5).

How diHerent are the results at other labeling times? Showing uptake curves (with errors) for more than just two peptides in the supplement Figure S12 might be helpful.

We found a continuous increase in deuterium uptake as we increased the exchange time from 0.5 to 240 min, which reached saturation at 120 min. Therefore, the exchange follows the same pattern at all time points. Butterfly plots at diNerent HDX times of 0.5 to 240 min are shown in gradient of light blue to dark blue which clearly shows the pattern of deuterium uptake at increasing incubation times (Figure 5). The HDX uptake kinetics of selected peptides with corresponding error bars are shown in Figure S12.

How redundant are the data, i.e. how good is the peptide coverage/resolution in key regions at the domain-domain interface that the authors deem important? Mapping the maximal deuterium uptake on the structures in Figure 5 is not very helpful. Perhaps mapping the whole range of uptake using a gradient color scheme would be more informative.

Overall coverage and redundancy for all four proteins are> 90% and > 4.0, respectively, with an average error margin in fractional uptake among all peptides is 0.04-0.05 Da, which suggests that our data is reliable (**Table S3**). We modified the main panel figures showing the gradient of deuterium uptake in blue-white-red for 0 to 30% of deuterium uptake on the chain A of the dimeric LCs.

(3) Is the conformational heterogeneity depicted in M&M simulations consistent with HDX results? The authors may want to address this by looking at the EX1/EX2 exchange kinetics for AL vs. non-AL proteins. Do AL proteins show more EX1?

No, we don’t see any EX1 exchange kinetics in our analysis. This is compatible with the prediction of the H-state that is a native like state and not an unfolded/partially folded state.

(4) Perhaps the main conclusion could be softened given the small number of proteins (six), esp. since only four (3 AL and 1 non-AL) could be explored by HDX. Are other HDX MS data of AL LCs from the same Lambda6 family (e.g. PMID: 34678302) consistent with the conclusions that a particular domain-domain interface is weakened in AL vs. non-AL LCs?

We thank the reviewer for this suggestions. A diNerence in HDX MS data is indeed visible between AL and MM proteins for peptide 33-47 in the suggested paper (Figures 4, S5 and S8). The diNerence is reduced by the mutation identified in the paper as driving the aggregation in that specific case. We now mention this in the discussion.

(5) Please clarify if the H* state is the same for a covalent vs. non-covalent LC dimer.

We do not know because our data are only for covalent dimers. But, interestingly, the state is very similar to what was observed for a model kappa light-chain in Weber, et al., we have better highlighted this point in the discussion.

(6) Please try and better explain why a smaller distance between CL domains in H7 protein and a larger distance in other AL proteins both promote protein misfolding.

We do not have elements to discuss this point in more detail.

(7) Please comment on the Kratky plots data vs. model agreement (see comments above).

Done.

(8) Please find a better way to display, describe, and interpret the HD exchange MS data.

We have generated new main text (new Figure 5) and SI figures that we think allow the reader to better appreciated our observations. Corresponding results sections have been also improved.

Minor points:(9) Is the population of the H-state with perturbed CL-CL domain interface, which was obtained in M&M simulations, suHicient to be observable by HDX MS?

While populations alone are not enough to determine what is observable by HDX MS, a 10% population correspond roughly to 6 kJ/mol of ΔG and is compatible with EX2 kinetics. Previous works suggested that HDX-MS data should be sensitive to subpopulations of the order of 10%, (https://doi.org/10.1016/j.bpj.2020.02.005, https://doi.org/10.1021/jacs.2c06148)

(10) Typically, an excited intermediate in protein unfolding is a monomer, while here it is an LC dimer. Is this unusual?

This is a good point, we think that intermediates have mostly been studied on monomeric proteins because these are more commonly used as model systems, but we do not feel like discussing this point.

(11) Low deuterium uptake is consistent with a rigid structure but may also reflect buried structure and/or structure that moves on a time scale greater than the labeling time.

We agree.

**Reviewer #3 (Recommendations for the authors):**
(1) The p-value (statistical significance) of Rg diHerence should be computed.

We thank the reviewer for the suggestion, we calculated the p-value that resulted quite significant.

(2) The significance of mutations (SHM?) at the interface, such as A40G should be compared with previous observations. (Garrofalo et al., 2021).

We thank the reviewer for the suggestion, a sentence has been added in the discussion.